# Perceived Neighborhood Environment and Its Association with Health Screening and Exercise Participation amongst Low-Income Public Rental Flat Residents in Singapore

**DOI:** 10.3390/ijerph16081384

**Published:** 2019-04-17

**Authors:** Liang En Wee, Yun Ying Tammy Tsang, Sook Muay Tay, Andre Cheah, Mark Puhaindran, Jaime Yee, Shannon Lee, Kellynn Oen, Choon Huat Gerald Koh

**Affiliations:** 1Singhealth Internal Medicine, Singapore General Hospital, Singapore 169608, Singapore; 2Duke-NUS Graduate Medical School, Singapore 169857, Singapore; 3Institute of Mental Health, Singapore 539747, Singapore; tammytyy@gmail.com; 4Department of Anesthesia, Singapore General Hospital, Singapore 169608, Singapore; tay.sook.muay@sgh.com.sg; 5Department of Hand and Reconstructive Surgery, National University Hospital, National University Health System, National University of Singapore, Singapore 119228, Singapore; andre_cheah@nuhs.edu.sg (A.C.); mark_e_puhaindran@nuhs.edu.sg (M.P.); 6Yong Loo Lin School of Medicine, National University Health System, National University of Singapore, Singapore 119228, Singapore; jaimeyee96@gmail.com (J.Y.); shannon.lxj@gmail.com (S.L.); kellynnoenqixuan@gmail.com (K.O.); 7Saw Swee Hock School of Public Health, National University of Singapore, National University Health System, Singapore 117549, Singapore; ephkohch@nus.edu.sg

**Keywords:** neighborhood environment, public housing, Asian, health behaviors

## Abstract

*Background*: In Singapore, an Asian city-state, more than 80% live in public housing. While the majority (90%) own their homes, a needy minority lives in rental flats. Public rental flats are built in the same location as owner-occupied blocks. We evaluated factors associated with perceptions of the neighborhood environment and its association with exercise and health screening participation. *Methods*: Logistic regression was used to identify associations between perceptions of the neighborhood environment (overall perceived neighborhood disadvantage, safety, and convenience) and sociodemographic factors, as well as exercise and screening participation, amongst residents aged ≥60 years in two Singaporean public housing precincts in 2016. *Results*: Our response rate was 62.1% (528/800). Staying in a rental flat independently was associated with increased neighborhood disadvantage (adjusted odds ratio, aOR = 1.58, 95%CI = 1.06–2.35). Staying in a stand-alone block (as opposed to staying in a mixed block comprised of both rental and owner-occupied units) was associated with perceptions of a poorer physical environment (aOR = 1.81, 95%CI = 1.22–2.68) and lower perceived proximity to recreational areas (aOR = 1.14, 95%CI = 1.04–1.25). Perceptions of neighborhood disadvantage were independently associated with reduced exercise participation (aOR = 0.67, 95%CI = 0.45–0.98) and reduced participation in diabetes screening (aOR = 0.63, 95%CI = 0.41–0.95). *Conclusion*: Despite sharing the same built environment, differences in the perception of the neighborhood environment between low-socioeconomic status (SES) and high-SES communities persist. Perceived neighborhood disadvantage is associated with lower participation in regular exercise and diabetes screening.

## 1. Introduction

Health and place are inextricably intertwined. It is well known that the characteristics of the neighborhood environment have an impact on the health and well-being of residents residing in these communities [1,2,3]. The effect of neighborhood on health can be divided into two mechanisms: compositional and contextual [4,5]. Compositional measurements aggregate individual socioeconomic characteristics in defined geographical units to serve as measurements of the community, while contextual measurements target the features of the built environment (the physical environment of the community) and social environment (e.g., community networks) [5]. Contextual factors, in turn, can be measured by both subjective and objective means. Objective measurements generally rely on neighborhood-level census tract data to quantify certain neighborhood characteristics, whereas subjective measurements depend on some element of perceived neighborhood quality [6]. While subjective and objective constructs of neighborhood quality are both related to individual health, perceived neighborhood quality is more strongly associated [6,7]. The majority of these studies, though, have been conducted in Western societies; only in recent years have there been studies from urban Asian societies that explore the link between neighborhood characteristics and health [8,9]. 

Singapore is one such example of a rapidly-urbanizing multi-ethnic Asian society. Home ownership is a key local indicator of socioeconomic status (SES) in Singapore. The majority of Singaporeans (≥ 85%) [10,11] live in public housing, and home ownership rates are high (87.2%) [10]. For the needy (<5% of population) who cannot afford their own home, heavily-subsidized public rental housing is available [12]. A unique characteristic is that in Singapore, public rental housing blocks are built within the same locations as owner-occupied public housing apartments. Studies of the neighborhood environment focus on the mesoscale (roughly taken to be the environment within walking distance of one’s home) [9,13]. In Singapore, due to geographic proximity, residents of both rental and owner-occupied public housing share the same built environment at the mesoscale, with rental flats either existing in stand-alone tower blocks or being combined with owner-occupied units in integrated tower blocks [14]. At the microscale, we have individual apartment units; followed by the apartment block at the mesoscale; and finally, the precinct/new town at the macroscale. Maintenance of the common areas is also carried out by the same public authority (town councils), creating a homogenous built environment. The physical plans of public housing have been designed, via quota systems and various subsidy schemes, to integrate various income and racial groups and prevent the development of low-income or ethnic ghettos [14]. 

Despite the homogeneity of the built environment, staying in a public rental flat in Singapore has been correlated with poorer measures of physical and mental health, amongst adult Singaporeans; even after controlling for individual SES (e.g., individual employment status, education, being a recipient of financial aid). Residents of low-SES public rental flats had higher prevalence of poorer physical and mental health, poorer management of chronic disease, and reduced access to health services. In terms of physical health, staying in a low-SES community was associated with poorer hypertension management [15], as well as higher rates of chronic pain [16]. For mental well-being, staying in a low-SES rental flat neighborhood was associated with poorer cognitive function [17] and higher depression rates [18] among the elderly. Overall, residents in public rental flats had lower health-related quality of life (HRQoL) compared to their counterparts staying in owner-occupied housing [19]. This also had an impact on access to health services: residents in public rental flats had lower access to cardiovascular and cancer screening [20,21,22] and were less likely to seek treatment from medical professionals [23]. Residents of public rental flats also had higher readmission risk and increased utilization of hospital services [24]. Put together, these findings demonstrate the impact of neighborhood SES on health in Singapore. However, few studies have examined contextual measurements of the neighborhood environment and its association with health in the Singaporean setting. As such, this study investigated perceptions of the neighborhood environment and its association with health-seeking behaviors amongst public rental housing residents in Singapore. 

## 2. Methodology

### 2.1. Study Population

All residents aged ≥60 years in two public housing precincts in Singapore in September 2016 were surveyed. There was a total of 15 apartment blocks within the two precincts. Within the precincts, there were 2 mixed blocks consisting of predominantly owner-occupied housing with some rental flats integrated within the same block; there were also 6 stand-alone rental flat blocks and 7 stand-alone owner-occupied blocks. All blocks that make up a precinct are adjacent to each other and share common amenities built within the vicinity. The response rate was calculated based on a combination of census information and information from grassroots organizations as to the number of residents aged ≥60 years residing in the blocks.

### 2.2. Study Methodology

#### 2.2.1. Baseline Information

At baseline, information on residents’ sociodemographic characteristics and medical, functional, and social status was collected via interviewer-administered questionnaires in English, Chinese, and Malay. Standardized training for questionnaire administration was provided to interviewers by the study coauthors. Interviewers also assessed if residents were adherent to regular screening for cardiovascular disease (hypertension, diabetes, dyslipidemia) and collected self-reported data from residents regarding lifestyle behaviors. Interviewers were medical students who underwent standardized training prior to study commencement. Comorbidity burden was measured using the Charlson Comorbidity Index (CCMI) [25]. Functional status in basic activities of daily living (bADL) was also quantified using the Katz Index, while social isolation was quantified using the Lubben Social Network Score-6 (LSNS 6) [26]. Loneliness was quantified using the 3-item UCLA Loneliness Scale [27].) HRQoL was quantified using the EQ5D [19]. Regular screening was defined as adherence to guidelines on health screening recommended by the local Ministry of Health [28]. For those aged >40 years and of unknown hypertensive/diabetes mellitus/dyslipidemia status, a blood pressure check is recommended yearly, and fasting glucose and lipids are recommended every 3 years. For lifestyle behaviors, regular exercise was defined as participation in any form of sports or exercise for at least 20 minutes per occasion, for 3 or more days a week. The definitions were in line with those used in the nationwide National Health Survey [29]. 

#### 2.2.2. Perceptions of the Neighborhood Environment

Subjective measures of the neighborhood environment aim to assess perceived personal safety, physical convenience, and social cohesion within the neighborhood [30]. A modified version of the Neighborhood Environment Walkability Scale-Abbreviated (NEWS-A) was used to survey residents’ perceptions of the neighborhood environment. The NEWS-A was conceived of to provide an empirically-derived yet succinct measure of various aspects of the built environment related to walkability [31]. In its original form, the NEWS-A comprised a total of 64 items spread across 12 subscales. However, not all subscales were relevant to the local context. The NEWS has been previously modified for use in the Singaporean setting [32], preserving 8 subscales of residential density, land use mix (diversity), land use mix (access), street connectivity, infrastructure (places for walking and cycling), aesthetics (neighborhood surroundings), traffic safety, and safety from crime. It has previously been utilized in the Singaporean context to study the impact of the built environment on physical activity [32]; however, in a separate study, specific subscales of the NEWS were utilized in analyzing the impact of the built environment on cognitive impairment in a population of Singaporean community-dwelling elderly [33]. This study utilized the NEWS-A subscales of crime safety, land-use mix (access), and land-use mix (diversity), but omitted the subscales of residential density, aesthetics, infrastructure (walking and cycling), street connectivity, and traffic safety. The neighborhood density subscale was removed as this study solely focused on public housing estates, which were a homogenous mix of high-rise apartment blocks (no single-story or detached residences). The street connectivity, traffic safety, aesthetics, and infrastructure (walking/cycling) subscales were also removed because these subscales were associated more with physical activity rather than measures of health [32]. In a local study, land use mix (diversity) was the only factor associated with cognitive impairment, whereas the other subscales were not significantly associated [33]. In its final form, we utilized 17 items over 3 subscales (crime safety: 7 items; land use access: 2 items; land use diversity: 8 items). Crime safety and land use access were originally reported as a 4-point Likert scale; land use diversity as a 5-point Likert scale. Some of the items were reverse-coded. The responses on the various items were summated to form a total score, as per the scoring system utilized by the NEWS-A. The median total score was then used to dichotomize the results into “less disadvantaged neighborhood” and “more disadvantaged neighborhood”. About one-third of the study participants (31.6%, 167/528) felt that they lived in a more disadvantaged neighborhood. In this study, as the scale utilized to measure perceptions of the neighborhood environment was significantly modified from the original NEWS-A score, we proceeded to conduct additional factor analysis. Factor analysis of the 17 NEWS items utilized in our study derived a total of 3 principal components, which were summarized as “perceived safety and convenience”, “perceived physical environment”, and “perceived proximity to recreational areas”. Most of the subscales related to perceived neighborhood safety (e.g., perceived crime rate in the neighborhood, perceived safety when walking around) and perceived neighborhood convenience (availability of amenities, such as convenient access to shops, eating places, medical services, and recreational areas) loaded onto the first principal component, while subscales related to the physical environment (e.g., presence of litter, lighting/signage, physical barriers such as uneven ground or steps) loaded onto the second principal component, and perceived proximity to recreational areas loaded onto the last principal component. The results for each principal factor were dichotomized using the median result as the cut-off. The Cronbach-alpha coefficient for the subscale items of the modified score was 0.71, suggesting reasonable internal consistency.

### 2.3. Statistical Analysis

Descriptive statistics were computed for the study population. Factor analysis of the 17 items (from the NEWS-A subscales of crime safety, land use access, and land use diversity) utilized in our study was conducted; the cut-off was set at *e* ≥ 1 based on an inspection of the scree plot. The Chi-square was used to identify associations between perception of neighborhood disadvantage and sociodemographic characteristics, as well as health-seeking behaviors, on univariate analysis; and logistic regression for multivariate analysis. For multivariate analysis, the most parsimonious logistic regression model was constructed by using the criterion of a *p*-value <0.1 on univariate analysis as a cut-off for entry of factors into the final multivariate model; and removing non-significant variables in a stepwise fashion till the most parsimonious model was achieved. Variables included in final multivariate logistic regression models were also evaluated for multicollinearity, with a view toward removing redundant variables; however, no multicollinearity was detected. All statistical analysis was performed using SPSS (Version 17.0, SPSS Inc, Chicago, IL, USA) and STATA (Version 22.0, StataCorp, Texas Station, NV, USA), and statistical significance was set at *p* < 0.05. 

### 2.4. Ethics Approval

Ethics approval was obtained from the National University of Singapore Institutional Review Board; informed consent was sought, and participation was voluntary.

## 3. Results

A total of 528 residents participated in the study. The response rate was 62.1% (528/800). About half (54.4%, 287/528) of the study population were aged ≥75 years; 70.1% (370/528) had only secondary education and below; and one third (33.9%, 179/528) had a household income of <S$1500/month, compared against the mean household income of S$8,800 in 2016 [34]. The median duration of residence in the neighborhood was eight years (interquartile ratio, IQR = 5–20). Out of 528 residents, 234 were staying in mixed blocks (owner-occupied and rental units in the same block) and 200 were staying in stand-alone blocks. Of those staying in rental flats within mixed blocks, 40.5% (15/37) perceived their neighborhood as more disadvantaged. Of those staying in stand-alone rental flat blocks, 37.8% (90/238) perceived their neighborhood as more disadvantaged. Of those staying in owner-occupied housing, 24.5% (62/253) perceived their neighborhood as more disadvantaged. 

On factor analysis, the 17 items (initially obtained from the NEWS-A subscales of crime safety, land use access, and land use diversity) could be reduced to three principal components, accounting for 62% of the variance (Appendix A). Twelve of the items loaded onto the first principal component, which was termed “perceived neighborhood safety and convenience”, because items associated with perceived neighborhood safety and crime rate, as well as items associated with perceived proximity to various amenities (e.g., grocery stores, public transport, medical services) loaded onto this principal component, accounting for 45.2% of the variance (eigenvalue, *e* = 7.69). Another four of the items loaded onto the second principal component, which was termed “physical environment”; this comprised items describing the physical environment, such as the presence of trash and good lighting/signage within the common areas. This factor accounted for 10.60% of the variance (*e* = 1.80). Finally, perceived proximity to recreational areas (e.g., parks and playgrounds) accounted for the third and last principal component, accounting for 6.16% of the variance (*e* = 1.05). 

The factors associated with perceptions of the neighborhood environment on univariate analysis are illustrated in Table 1. Overall, on univariate analysis, staying in a stand-alone block, staying in the neighborhood for a longer duration (>8 years), staying in a rental flat, staying in three-room apartments or smaller, being single, not having a religion, aged >75 years, having ≥3 people in the same household, having more medical comorbidities, requiring a caregiver, being socially isolated, and loneliness were associated with perceived neighborhood disadvantage (*p* < 0.05). 

The factors associated with perceived neighborhood safety and convenience, physical environment, and proximity to recreational areas, on univariate analysis, are illustrated in Table 2. Staying in the neighborhood for a longer duration (>8 years), staying in a rental flat, being single, not having a religion, having a lower household income, being socially isolated, and experiencing loneliness were associated with lower perceived neighborhood safety and convenience. Living in a stand-alone block (as opposed to living in a mixed block), living in a rental flat, being female, and having no caregiver were associated with poorer perceived physical environment. Living in a stand-alone block, staying in the neighborhood for >8 years, staying in a rental flat, staying in three-room apartments or smaller, aged >60 years, lower household income, and having more medical comorbidities were associated with poorer perceived proximity to recreational areas.

The factors associated with perceptions of the neighborhood environment on multivariate analysis are illustrated in Table 3. On multivariate analysis, staying in a rental flat apartment was independently associated with increased perceptions of neighborhood disadvantage (adjusted odds ratio, aOR = 1.58, 95%CI = 1.06–2.35). Marital status and social isolation were also independent predictors of overall perceptions of neighborhood disadvantage. Staying in a stand-alone block (as opposed to staying in a mixed block comprised of both rental and owner-occupied units) was associated with perceptions of a poorer physical environment (aOR = 1.81, 95%CI = 1.22–2.68); being of female gender and having no caregiver were also independently associated with perceptions of a poorer physical environment. Staying in a stand-alone block was also independently associated with lower perceived proximity to recreational areas (aOR = 1.14, 95%CI = 1.04–1.25), together with older individual age (aOR = 1.64, 95%CI = 1.14–2.36).

The association between perceptions of the neighborhood environment and regular health screening participation and regular exercise are illustrated in Table 4 (univariate analysis) and Table 5 (multivariate analysis). Perceptions of overall neighborhood disadvantage were independently associated with reduced exercise participation (aOR = 0.67, 95% CI = 0.45–0.98), as well as reduced participation in regular diabetes screening (aOR = 0.63, 95%CI = 0.41–0.95). 

## 4. Discussion

Housing is an important social determinant of health in Singapore. Previous studies in public housing in Singapore showed that perceived neighborhood safety and diversity of amenities are key components of residents’ subjective perceptions of neighborhood quality [32]. Despite efforts at planned integration of lower-SES rental blocks in the same shared environment as owner-occupied blocks, staying in a rental flat was independently associated with differences in perceived neighborhood disadvantage and neighborhood safety, as well as convenience. Several factors may account for this. Despite relative homogeneity of the built environment between lower-SES rental blocks and higher-SES owner-occupied blocks, subtle differences in the physical environment can still exist. Sociological observations suggest that though the physical façade of rental and non-rental blocks may be similar, communal areas in rental blocks may not be as well maintained, resulting in subtle differences in the neighborhood environment such as the sight of trash and lingering odors in common areas of the rental blocks [35]. Additionally, signage and posters warning against criminal activity are more common in rental blocks and may contribute to a strong sense of negativity and foster a sense of insecurity, distrust, and danger [35], reducing perceptions of neighborhood safety. With regards to perceived neighborhood convenience, differences persisted even though rental and non-rental residents live in the same geographic location (and hence, distances to amenities might be expected to be similar). Tensions do exist between residents of owner-occupied blocks and rental blocks when it comes to the use of shared amenities [35], and this may reflect in subjective perceptions of diminished access and perceived convenience (even though objective measures of access, such as geographic proximity, are similar). Finally, while residents of rental and non-rental flats may share the same built environment, this does not translate into sharing the same social environment. Greater turnover in the rental flat population can diminish the potential for forming social interactions that improve neighborhood cohesion and perceptions of neighborhood safety. Although rental flats are built in the same precincts as their owner-occupied counterparts, residents of different blocks may not meet. Studies have demonstrated that much of the social interaction amongst residents of public housing in Singapore occurs within the block, with only occasional interactions between residents from different blocks [36]. Hence, residents in rental housing may also be stigmatized [35,37] and tend to keep to themselves, perceiving their neighborhood environment as more disadvantaged even though the built environment is homogenous. 

Staying in a stand-alone block (as opposed to staying in a mixed block comprised of both rental and owner-occupied units) was associated with perceptions of a poorer living environment and reduced proximity to recreational areas. Perhaps in mixed blocks, demarcation between those of lower-SES (rental flat dwellers) and higher-SES (those who own their own unit) is less stark, whereas in stand-alone blocks where there are clear demarcations between rental and owner-occupied blocks, disparities are accentuated. With clear demarcations between stand-alone rental and owner-occupied blocks, perhaps residents in the rental flat blocks are constantly reminded of the differences in their social standing and hence perceive their neighborhood to be more deprived [35,38]. On the other hand, residents in stand-alone owner-occupied blocks feel that the presence of clearly-demarcated stand-alone rental blocks in their midst results in a disamenity [39] and hence also perceive their neighborhood as more deprived. Conversely, in mixed blocks, blurring of the divide between rental and owner-occupied flats occurs, and these perceived inequalities are not as obvious. Again, tensions that exist when recreational areas need to be shared may be more marked when the boundaries between stand-alone owner-occupied blocks and rental blocks are clearly drawn, resulting in subjective perceptions of diminished access, whereas in the anonymity of a mixed block, these tensions may subside. 

Perceptions of the neighborhood, in turn, are associated with exercise participation and participation in regular diabetes screening. It is well known that perceptions of neighborhood walkability influence physical activity; [40] neighborhood security is also an important consideration influencing usage of recreational facilities [41,42]. In this study, perceptions of overall neighborhood disadvantage were independently associated with reduced exercise participation. This was in keeping with other local studies demonstrating that diversity of land use mix and close proximity to amenities and facilities were associated with higher frequency of walking for transportation purposes [32]. Apart from the physical characteristics of the neighborhood environment that may promote walkability, better perceived neighborhood quality may also promote an improved sense of purpose and meaning in life, which may predispose individuals to take control of their own health, such as increasing exercise frequency and physical activity [42]. Poorer perceived neighborhood quality may also contribute to increased loneliness amongst community-dwelling elderly [43]; which can reduce physical activity, given the importance of social presence in encouraging physical activity amongst the elderly [44]. Given the association between perceived neighborhood disadvantage and residence in a rental flat block in our population, facilities for recreation in proximity to rental flat blocks can be enhanced in order to encourage exercise participation. There is less data, however, on the association between perceived neighborhood quality and health screening participation, with a lack of previous studies investigating such an association in the published literature [45]. A 2016 study demonstrated an association between perceptions of social and physical disorder, such as fear of crime and visible garbage, and cancer screening rates [46]. Various reasons were postulated to support this association; perhaps the inherent disorder in lower-SES environments contributes to feelings of powerlessness and inaction regarding individuals’ own health, or the lack of social support and normative behaviors regarding screening in lower-SES environments contributes to low screening rates [47]. In our study, although there was no significant association between perceived neighborhood quality and gynecological cancer screening, we demonstrated an association between perceived poorer neighborhood quality and lower odds of regular diabetes screening participation amongst non-diabetics. There were lower proportions of residents participating in regular dyslipidemia screening amongst those staying in areas with poorer perceived neighborhood quality; conversely, higher proportions of residents staying in areas with poorer perceived neighborhood quality participated in regular hypertension screening, although in both cases, the association was not statistically significant. In Singapore, while residents in rental flat blocks have lower rates of participation in regular cancer and cardiovascular screening [21,22], there are significant differences in accessibility between screening modalities. In general, cardiovascular screening takes place at either private general practitioners (GPs) or public primary care clinics (“polyclinics”) in the immediate neighborhood; [21] these clinics are distributed at relatively high densities, with at least one clinic within a 0.5-kilometer radius of the residential block. [23] However, in the case of cancer screening, subsidized screening is generally available only at designated polyclinics with the necessary facilities (e.g., imaging) [22]; hence, the facilities providing subsidized cancer screening are generally located outside of the immediate neighborhood. This may account for why we detected a significant association for diabetes screening, but not cancer screening; as diabetes screening occurs at centers within the immediate neighborhood, perceived neighborhood quality does impact perceived accessibility and hence the diabetes screening participation, whereas cancer screening, occurring outside the immediate neighborhood, is less impacted by the perceived quality of the immediate neighborhood. Hypertension screening, unlike diabetes screening, was higher in neighborhoods with lower perceived neighborhood quality, though the association was not statistically significant. Though hypertension management tends to be poorer in rental flat blocks, with prioritization of patients staying in rental housing for various interventions to improve chronic disease management [15], with wider availability of home-based blood pressure monitoring and more widespread provision of blood pressure screening in door-to-door screening [15], residents may no longer be so dependent on primary care clinics in the neighborhood for hypertension screening, compared with diabetes screening, which still requires phlebotomy equipment for blood draws and a follow-up system to remind the patient to review results. Given the known correlation between the built environment and various cardiovascular disease outcomes [48,49], poorer-quality neighborhoods can be prioritized in cardiovascular screening interventions in order to facilitate early detection and mitigate the risk of cardiovascular disease [21]. 

The limitations of our study were as follows. As this was a cross-sectional study, not a prospective one, we can only identify correlation, but not causation. Additionally, this study was carried out in two geographic sites; we were unable to obtain a nationally-representative sample of the rental flat population in Singapore because of logistical difficulties, as rental flats are scattered across the entire country in socially-integrated precincts. However, we note that our study population was fairly similar in terms of sociodemographic makeup when compared against national data on low-income neighborhoods [11]. In our study, we utilized subjective, not objective, measures of neighborhood environment. However, in our local context, subjective measures of neighborhood environment were more closely associated with mental and physical health [9]. Finally, we elected to use subscales of the NEWS-A in our assessment of perceived neighborhood quality that were relevant to our study context, similar to previous studies in our local population that also utilized selected elements of the NEWS-A [32,33]; the rationale was that previous local studies could serve as a potential basis of comparison, given the extremely limited number of local studies that have evaluated neighborhood perceptions. However, we acknowledge that partial usage of the NEWS-A questionnaire, which was specifically designed to evaluate perceptions of neighborhood walkability, may potentially affect the validity of our measurements; elements of perceived neighborhood quality identified in this study (safety and convenience, physical environment) may form the basis for future constructs designed to assess neighborhood perceptions in our local population specifically. 

## 5. Conclusions

Even though needy residents share the same built environment compared to their more well-to-do neighbors in densely urbanized Singapore, there are still differences in perception of the neighborhood environment. These differences may be due to subtle differences in the physical environment, as well as differences in the social environment. Staying in a stand-alone block, as opposed to staying in an integrated block comprised of both rental and owner-occupied units, was associated with perceptions of a poorer living environment and reduced proximity to recreational areas; perhaps because clear demarcations between rental and owner-occupied blocks may result in increased perceived inequality. Having a poorer perception of the neighborhood environment is associated with reduced participation in regular exercise and diabetes screening.

## Figures and Tables

**Table 1 ijerph-16-01384-t001:** Associations between overall neighborhood perception and geographical, sociodemographic, medical, and social factors, amongst residents in two public housing precincts in Singapore, on univariate analysis (*n* = 528).

Overall Neighborhood Perception			
Geographical, Sociodemographic, Medical, and Social Factors	Less Disadvantaged (*n* = 361) *n* (%) ^1^	More Disadvantaged (*n* = 167) (*n* %) ^1^	OR (95% CI)
Geographical			
Site			
Staying in a mixed block	171 (73.1)	63 (26.9)	1.00
Staying in a stand-alone block	190 (64.6)	10 (35.4)	1.49 (1.02–2.16) *
Stayed in neighborhood for >8 years			
No	223 (72.9)	83 (27.1)	1.00
Yes	138 (62.2)	84 (37.8)	1.64 (1.13–2.37) *
Staying in rental apartment vs. owner-occupied			
Owner-occupied	191 (75.8)	61 (24.2)	1.00
Rental	170 (61.6)	106 (38.4)	1.95 (1.34–2.85) ***
Number of rooms			
3 rooms or smaller	240 (65.4)	127 (34.6)	1.00
4–5 rooms	121 (75.2)	40 (24.8)	0.63 (0.41–0.95) *
Socio-demographic			
Gender			
Female	211 (68.5)	97 (31.5)	1.00
Male	150 (68.2)	70 (31.8)	1.02 (0.70–1.47)
Marital status			
Not married	141 (58.8)	99 (41.2)	1.00
Married	220 (76.4)	68 (23.6)	0.44 (0.30–0.64) ***
Religious			
No	101 (60.8)	65 (39.2)	1.00
Yes	260 (71.8)	102 (28.2)	0.61 (0.41–0.90) *
Age			
Age 60–75 years	175 (72.6)	66 (27.4)	1.00
Age ≥75 years	186 (64.8)	101 (35.2)	1.14 (0.99–2.09)
Currently employed			
No	192 (66.2)	98 (33.8)	1.00
Yes	169 (71.0)	69 (29.0)	0.80 (0.55–1.16)
Education			
Secondary and below	109 (69.0)	49 (31.0)	1.00
Post-secondary and above	252 (68.1)	118 (31.9)	1.04 (0.70–1.56)
Number of people in household			
2 or less people	211 (72.0)	82 (28.0)	1.00
3 or more people	150 (63.8)	85 (36.2)	1.46 (1.01–2.11) *
Average household income			
≤$1500/month	229 (65.6)	120 (34.4)	1.00
>$1500/month	132 (73.7)	47 (26.3)	0.68 (0.46–1.01)
Medical and functional status			
Comorbidity (Charlson Comorbidity Index)			
CCMI = 0	280 (71.1)	114 (28.9)	1.00
CCMI ≥ 1	81 (60.4)	53 (39.6)	1.61 (1.07–2.42) *
Chronic pain (pain >6 months)			
No	318 (67.9)	150 (32.1)	1.00
Yes	43 (71.7)	17 (28.3)	0.84 (0.46–1.52)
Anxiety/mood issues			
No	335 (68.4)	155 (31.6)	1.00
Yes	26 (68.4)	12 (31.6)	1.00 (0.49–2.03)
Functional status (basic activities of daily living)			
Dependent in at least 1 bADL	9 (52.9)	8 (47.1)	1.00
Independent in all bADLs	352 (68.9)	159 (31.1)	0.51 (0.19–1.34)
Social network			
Has caregiver			
No	297 (70.7)	123 (29.3)	1.00
Yes	64 (59.3)	44 (40.7)	1.66 (1.08–2.57) *
Social isolation (Lubben Social Network Score-6)			
No (LSNS < 12)	166 (77.9)	47 (22.1)	1.00
Yes (LSNS ≥ 12)	195 (61.9)	120 (38.1)	2.17 (1.46–3.23) ***
Loneliness (UCLA Loneliness Scale)			
No (<6)	288 (71.8)	113 (28.2)	1.00
Yes (≥6)	73 (57.5)	54 (42.5)	1.89 (1.25–2.85) **

* = *p* < 0.05; ** *p* < 0.01; *** = *p* < 0.001; ^1^ the median total score derived from the Neighborhood Environment Walkability Scale-Abbreviated (NEWS-A) subscales of crime safety, land use access, and land use diversity (with a maximum score of 76 and a minimum score of 17) was used as a cut-off to dichotomize into “overall neighborhood perception: less disadvantaged” and “overall neighborhood perception: more disadvantaged”. The median score was 40 (interquartile ratio = 28–60).

**Table 2 ijerph-16-01384-t002:** Associations between perceived neighborhood safety and convenience, perceived physical environment, and perceived proximity to recreational areas, amongst residents in two public housing precincts in Singapore, on univariate analysis (*n* = 528).

	Principal Components Analysis (Perceived Safety and Convenience; Perceived Physical Environment; Perceived Proximity to Recreational Areas)					
Geographical, Sociodemographic, Medical, and Social Factors	Less Safe and Convenient (*n* = 248) ^1^	OR (95% CI)	Poorer Physical Environment (*n* = 162) ^1^	OR (95% CI)	Lower Perceived Proximity to Recreational Areas (*n* = 203) ^1^	OR (95% CI)
Geographical						
Site						
Staying in a mixed block	102 (43.6)	1.00	57 (24.4)	1.00	72 (30.8)	1.00
Staying in a stand-alone block	146 (49.7)	1.12 (0.96–1.31)	105 (35.7)	1.73 (1.18–2.53) **	131 (44.6)	1.80 (1.26–2.59) **
Stayed in neighborhood for >8 years						
No	130 (42.5)	1.00	84 (27.5)	1.00	101 (33.0)	1.00
Yes	118 (53.2)	1.23 (1.04–1.46) *	78 (35.1)	1.43 (0.99–2.08)	102 (45.9)	1.73 (1.21–2.46) **
Staying in rental apartment vs. owner-occupied						
Owner-occupied	101 (40.1)	1.00	65 (25.8)	1.00	86 (34.1)	1.00
Rental	147 (53.3)	1.28 (1.09–1.51) **	97 (35.1)	1.56 (1.07–2.27) *	117 (42.4)	1.42 (1.01–2.02) *
Number of rooms						
3 rooms or smaller	180 (49.0)	1.00	118 (32.2)	1.00	154 (42.0)	1.00
4–5 rooms	68 (42.2)	0.88 (0.75–1.04)	44 (27.3)	0.79 (0.53–1.20)	49 (30.4)	0.61 (0.41–0.90) *
Socio-demographic						
Gender						
Female	145 (47.1)	1.00	105 (34.1)	1.00	124 (40.3)	1.00
Male	103 (46.8)	1.00 (0.85–1.17)	57 (25.9)	0.68 (0.46–0.99) *	79 (35.9)	0.83 (0.58–1.20)
Marital status						
Not married	132 (55.0)	1.00	74 (30.8)	1.00	97 (40.4)	1.00
Married	116 (40.3)	0.75 (0.64–0.89) ***	88 (30.6)	0.99 (0.68–1.43)	106 (36.8)	0.86 (0.60–1.22)
Religious						
No	93 (56.0)	1.00	57 (34.3)	1.00	71 (42.8)	1.00
Yes	155 (42.8)	0.77 (0.63–0.93) **	105 (29.0)	0.78 (0.53–1.16)	132 (36.5)	0.77 (0.53–1.12)
Age						
Age 60–75 years	105 (43.6)	1.00	78 (32.4)	1.00	75 (31.1)	1.00
Age ≥75 years	143 (49.8)	1.12 (0.96–1.32)	84 (29.3)	0.87 (0.60–1.25)	128 (44.6)	1.78 (1.25–2.55) **
Currently employed						
No	143 (49.3)	1.00	82 (28.3)	1.00	120 (41.4)	1.00
Yes	105 (44.1)	0.91 (0.77–1.07)	80 (33.6)	1.28 (0.89–1.86)	83 (34.9)	76 (0.53–1.08)
Education						
Secondary and below	79 (50.0)	1.00	48 (30.4)	1.00	68 (43.0)	1.00
Post-secondary and above	169 (45.7)	0.92 (0.78–1.10)	114 (30.8)	1.02 (0.68–1.53)	135 (36.5)	0.76 (0.52–1.11)
Number of people in household						
2 or less people	131 (44.7)	1.00	88 (30.0)	1.00	105 (35.8)	1.00
3 or more people	117 (49.8)	1.10 (0.94–1.30)	74 (31.5)	1.08 (0.74–1.55)	98 (41.7)	1.28 (0.90–1.82)
Average household income						
≤$1500/month	176 (50.4)	1.00	111 (31.8)	1.00	148 (42.4)	1.00
>$1500/month	72 (40.2)	0.83 (0.71–0.97) *	51 (28.5)	0.85 (0.58–1.27)	55 (30.7)	0.60 (0.41–0.88) *
Medical and functional status						
Comorbidity (Charlson Comorbidity Index)						
CCMI = 0	179 (45.4)	1.00	125 (31.7)	1.00	141 (35.8)	1.00
CCMI ≥ 1	69 (51.5)	1.12 (0.92–1.37)	37 (27.6)	0.82 (0.53–1.27)	62 (46.3)	1.55 (1.04–2.30) *
Chronic pain (pain >6 months)						
No	220 (47.0)	1.00	141 (30.1)	1.00	173 (37.0)	1.00
Yes	28 (46.7)	0.99 (0.77–1.28)	21 (35.0)	1.25 (0.71–2.20)	30 (50.0)	1.71 (0.99–2.93)
Anxiety/mood issues						
No	227 (46.3)	1.00	147 (30.0)	1.00	190 (38.8)	1.00
Yes	21 (55.3)	1.20 (0.84–1.73)	15 (39.5)	1.52 (0.77–3.00)	13 (34.2)	0.82 (0.41–1.64)
Functional status (basic activities of daily living)						
Dependent in at least 1 bADL	11 (64.7)	1.00	5 (29.4)	1.00	9 (52.9)	1.00
Independent in all bADLs	237 (46.4)	0.66 (0.34–1.26)	157 (30.7)	1.07 (0.37–3.07)	194 (38.0)	0.54 (0.21–1.43)
Social network						
Has caregiver						
No	190 (45.2)	1.00	138 (32.9)	1.00	164 (39.0)	1.00
Yes	58 (53.7)	1.18 (0.95–1.48)	24 (22.2)	0.58 (0.36–0.96) *	39 (36.1)	0.88 (0.57–1.37)
Social isolation (Lubben Social Network Score-6)						
No (LSNS < 12)	82 (38.5)	1.00	73 (34.3)	1.00	72 (33.8)	1.00
Yes (LSNS ≥ 12)	166 (52.7)	1.30 (1.11–1.52) ***	89 (28.3)	0.76 (0.52–1.10)	131 (41.6)	1.39 (0.97–2.00)
Loneliness (UCLA Loneliness Scale)						
No (<6)	175 (43.6)	1.00	132 (32.9)	1.00	150 (37.4)	1.00
Yes (≥6)	73 (57.5)	1.33 (1.07–1.65) **	30 (23.6)	0.63 (0.40–1.00)	53 (41.7)	1.20 (0.80–1.80)

* = *p* < 0.05; ** = *p* < 0.01; *** = *p* < 0.001; ^1^ factor analysis of the 17 NEWS items from the subscales of crime safety, land use access, and land use diversity utilized in our study derived a total of 3 principal components, which were summarized as “perceived safety and convenience”, “perceived physical environment”, and “perceived proximity to recreational areas”. The component subscales for each principal component were summated, and the results for each principal factor were dichotomized using the median result as the cut-off. The median score for “perceived safety and convenience” was 30 (min = 12, max = 45). The median score for “perceived physical environment” was 20 (min = 4, max = 16). The median score for “perceived proximity for recreational areas” was 3 (min = 1, max = 5).

**Table 3 ijerph-16-01384-t003:** Associations between perception of neighborhood environment and geographical, sociodemographic, medical and social factors, amongst residents in two public housing precincts in Singapore, on multivariate analysis (*n* = 528).

Overall Neighborhood Perception		
Perception of Neighborhood Environment as More Disadvantaged	Adjusted Odds Ratio, aOR (95% CI) ^1,2^	*p*-Value
Staying in rental apartment block vs. owner-occupied apartment block		
Owner-occupied	1.00	0.024
Rental apartment	1.58 (1.06–2.35)	
Marital status		
Not married	1.00	<0.001
Married	0.49 (0.33–0.73)	
Social isolation (Lubben Social Network Score-6)		
No (LSNS < 12)	1.00	<0.001
Yes (LSNS ≥ 12)	2.04 (1.36–3.01)	
Perceived neighborhood safety; physical living environment; proximity to recreational areas		
Perception of neighborhood environment as less safe and convenient	Adjusted odds ratio, aOR (95% CI) ^1,3^	*p*-value
Staying in rental apartment block vs. owner-occupied apartment block		
Owner-occupied	1.00	0.046
Rental apartment	1.44 (1.01–2.08)	
Marital status		
Not married	1.00	0.004
Married	0.60 (0.42–0.85)	
Social isolation (Lubben Social Network Score-6)		
No (LSNS < 12)	1.00	0.004
Yes (LSNS ≥ 12)	1.69(1.18–2.43)	
Perception of poorer physical environment	Adjusted odds ratio, aOR (95% CI) ^1,4^	*p*-value
Site		
Staying in mixed development	1.00	0.003
Staying in stand-alone block	1.81 (1.22–2.68)	
Gender		
Male	1.00	0.016
Female	1.61 (1.09–2.38)	
Has a caregiver		
No caregiver	1.00	0.030
Has a caregiver	0.57 (0.35–0.95)	
Perceived lower proximity to recreational areas	Adjusted odds ratio, aOR (95% CI) ^1,5^	*p*-value
Site		
Staying in mixed development	1.00	0.006
Staying in stand-alone block	1.14 (1.04–1.25)	
Age		
Age 60–75 years	1.00	0.008
Age ≥75 years	1.64 (1.14–2.36)	

^1^ The most parsimonious logistic regression model was constructed by using a criterion of a *p*-value <0.1 on univariate analysis as a cut-off for entry of factors into the final multivariate model; and removing non-significant variables in a stepwise fashion till the most parsimonious model was achieved. All variables significant on multivariate analysis are enumerated. ^2^ R^2^ of the final logistic regression model = 0.58. ^3^ R^2^ of the final logistic regression model = 0.53. ^4^ R^2^ of the final logistic regression model = 0.42. ^5^ R^2^ of the final logistic regression model = 0.39.

**Table 4 ijerph-16-01384-t004:** Associations between perception of neighborhood environment and regular screening participation and exercise, amongst residents in two public housing precincts in Singapore, on univariate analysis (*n* = 528).

Overall Neighborhood Perception				Principal Components Analysis (Perceived Safety and Convenience; Perceived Physical Environment; Perceived Proximity to Recreational Areas)					
Perception of Neighborhood Environment	Less Disadvantaged (*n* = 361) *n* (%)	More Disadvantaged (*n* = 167) (*n* %)	OR (95% CI)	Less Safe and Convenient (*n* = 248)	OR (95% CI)	Poorer Living Environment (*n* = 162)	OR (95% CI)	Lower Perceived Proximity to Recreational Areas (*n* = 203)	OR (95% CI)
Health screening participation									
Regular diabetes screening in non-diabetics									
Not going for regular screening	128 (64.6)	70 (35.4)	1.00	100 (50.5)	1.00	57 (28.8)	1.00	70 (35.4)	1.00
Going for regular screening	181 (73.9)	64 (26.1)	0.65 (0.43–0.97) *	104 (42.4)	0.86 (0.72–1.03)	80 (32.7)	1.20 (0.80–1.80)	93 (38.0)	1.12 (0.76–1.65)
Regular hyperlipidemia screening in non-dyslipidemics									
Not going for regular screening	112 (65.5)	59 (34.5)	1.00	85 (49.7)	1.00	44 (25.7)	1.00	61 (35.7)	1.00
Going for regular screening	115 (74.2)	40 (25.8)	0.66 (0.41–1.07)	64 (41.3)	0.86 (0.70–1.05)	55 (35.5)	1.59 (0.99–2.55)	49 (31.6)	0.83 (0.53–1.32)
Regular blood pressure screening in non-hypertensives									
Not going for regular screening	84 (74.3)	29 (25.7)	1.00	47 (41.6)	1.00	36 (31.9)	1.00	47 (41.6)	1.00
Going for regular screening	157 (69.2)	70 (30.8)	1.29 (0.78–2.15)	106 (46.7)	1.10 (0.90–1.34)	79 (34.8)	1.14 (0.71–1.85)	79 (34.8)	0.75 (0.47–1.19)
Regular pap smear in females									
Not going for regular screening	160 (65.8)	83 (34.2)	1.00	120 (49.4)	1.00	80 (32.9)	1.00	103 (42.4)	1.00
Going for regular screening	51 (78.5)	14 (21.5)	0.53 (0.28–1.01)	25 (38.5)	0.82 (0.65–1.03)	25 (38.5)	1.27 (0.72–2.25)	21 (32.3)	0.65 (0.36–1.16)
Regular mammogram in females aged 40–65 years									
Not going for regular screening	92 (75.4)	30 (24.6)	1.00	50 (41.0)	1.00	44 (36.1)	1.00	44 (36.1)	1.00
Going for regular screening	22 (71.0)	9 (29.0)	1.26 (0.52–3.02)	15 (48.4)	1.14 (0.80–1.67)	10 (32.3)	0.84 (0.36–1.95)	10 (32.3)	0.84 (0.37–1.95)
Exercise participation									
Exercise regularly (at least 30 mins, 5 or more times a week)									
No	168 (63.9)	95 (36.1)	1.00	139 (52.8)	1.00	77 (29.3)	1.00	107 (40.7)	1.00
Yes	193 (72.8)	72 (27.2)	0.66 (0.46–0.96) *	109 (41.1)	0.80 (0.68–0.94) **	85 (32.1)	1.14 (0.79–1.65)	96 (36.2)	0.83 (0.58–1.18)

* *p* < 0.05; ** *p* < 0.01

**Table 5 ijerph-16-01384-t005:** Associations between perception of neighborhood environment and regular diabetes screening participation and exercise, amongst residents in two public housing precincts in Singapore, on multivariate analysis (*n* = 528).

Participating in Regular Exercise (at Least 30 mins, 5 or More Times a Week)	Adjusted Odds Ratio, aOR (95% CI) ^1,2^	*p*-Value
Geographic Factors		
Perception of neighborhood environment		
Less disadvantaged	1.00	0.045
More disadvantaged	0.67 (0.45–0.98)	
Number of rooms		
3 rooms or smaller	1.00	0.012
4-5 rooms	1.67 (1.12–2.49)	
Demographic factors		
Age		
Age 60–75 years	1.00	<0.001
Age ≥75 years	1.91 (1.32–2.78)	
Medical factors		
Chronic pain (pain >6 months)		
No chronic pain	1.00	0.024
Has chronic pain	0.51 (0.29–0.92)	
Social factors		
Participate in community activities		
Not participating actively in community activities	1.00	0.005
Participating actively in community activities	1.71 (1.18–2.50)	
Social isolation (Lubben Social Network Score-6)		
No (LSNS < 12)	1.00	0.043
Yes (LSNS ≥ 12)	0.68 (0.47–0.98)	
Geographic factors		
Perception of neighborhood environment		
Less disadvantaged	1.00	0.027
More disadvantaged	0.63 (0.41–0.95)	
Medical factors		
On regular medical follow-up		
No	1.00	0.047
Yes	1.48 (1.01–2.18)	
Has dyslipidemia		
No	1.00	0.011
Yes	1.72 (1.13–2.62)	
In a state of perfect self-reported health (EQ5D)		
No	1.00	0.023
Yes	0.62 (0.41–0.94)	

^1^ The most parsimonious logistic regression model was constructed by using a criterion of a *p*-value <0.1 on univariate analysis as a cut-off for entry of factors into the final multivariate model; and removing non-significant variables in a stepwise fashion till the most parsimonious model was achieved. All variables significant on multivariate analysis are enumerated. ^2^ R^2^ of the final logistic regression model = 0.66. ^3^ R^2^ of the final logistic regression model = 0.57.

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
