# Peer review of "Perceived Neighborhood Environment and Its Association with Health Screening and Exercise Participation amongst Low-Income Public Rental Flat Residents in Singapore"

_ijerph, 2019, doi:10.3390/ijerph16081384_

Round 1
Reviewer 1 Report
Please specify which types of main component analyses were used to create the built environment index. The tables are a little difficult to understand, especially Table 1, it is no longer known what the reference value is (less disadvantaged and more disadvantaged). I suggest that Table 1 be divided into two parts. The first is on Overall neighbourhood perception and the other on Principal components analysis. Indicate what the reference value is. For multivariate analyses, did you include all dependent variables at once or did you proceed using a stepwise method? This must be indicated. The adjustment of odds ratio in Tables 2 and 4 are adjusted according to which variables, it must be indicated.
In the conclusion or discussion, it would be interesting to add a few sentences on possible public health actions to be taken to improve the situation and gaps.
Author Response
Reviewer 1:
Please specify which types of main component analyses were used to create the built environment index. The tables are a little difficult to understand, especially Table 1, it is no longer known what the reference value is (less disadvantaged and more disadvantaged). I suggest that Table 1 be divided into two parts. The first is on Overall neighbourhood perception and the other on Principal components analysis.
We thank the reviewer for the suggestion. We have divided the Table 1 as per Reviewer 1’s suggestion, for better clarity.
Indicate what the reference value is.
We have done so.
For Table 1, we include the following footnote:
“The median modified NEWS-A score (with a maximum score of 76 and a minimum score of 17) was used as a cutoff to dichotomise into “overall neighbourhood perception: less disadvantaged” and “overall neighbourhood perception: more disadvantaged”. The median score was 40 (interquartile ratio=28-60).”
For Table 2, we include the following footnote:
“Factor analysis of the 17 NEWS items utilised in our study derived a total of 3 principal components, which were summarised as “perceived safety and convenience”, “perceived physical environment”, and “perceived proximity to recreational areas”. The component subscales for each principal component were summated and the results for each principal factor were dichotomised using the median result as the cut-off. The median score for “perceived safety and convenience” was 30 (min=12, max=45). The median score for “perceived physical environment” was 20 (min=4, max=16). The median score for “perceived proximity for recreational areas” was 3 (min=1, max=5).”
For multivariate analyses, did you include all dependent variables at once or did you proceed using a stepwise method? This must be indicated.
We thank the reviewer for the query. We used a stepwise method. We have tried to increase the clarity by including the following lines in the Methods,
“For multivariate analysis, the most parsimonious logistic regression model was constructed by using a criterion of p-value<0.1 on univariate analysis as a cut-off for entry of factors into the final multivariate model; and removing non-significant variables in a stepwise fashion till the most parsimonious model was achieved.”
The adjustment of odds ratio in Tables 2 and 4 are adjusted according to which variables, it must be indicated.
We thank the reviewer for the query.
We have tried to increase the clarity by including the following footnote in Table 3 (previous Table 2) and Table 5 (previous Table 4).
“The most parsimonious logistic regression model was constructed by using a criterion of p-value<0.1 on univariate analysis as a cut-off for entry of factors into the final multivariate model; and removing non-significant variables in a stepwise fashion till the most parsimonious model was achieved. All variables significant on multivariate analysis are enumerated.”
In the conclusion or discussion, it would be interesting to add a few sentences on possible public health actions to be taken to improve the situation and gaps.
We thank the reviewer for the feedback. We have added the following lines to the Discussion:
“Apart from physical characteristics of the neighbourhood environment that may promote walkability, better perceived neighborhood quality may also promote an improved sense of purpose and meaning in life, which may predispose individuals to take control of their own health, such as increasing exercise frequency and physical activity. Poorer perceived neighbourhood quality may also contribute to increased loneliness amongst community-dwelling elderly; which can reduce physical activity, given the importance of social presence in encouraging physical activity amongst the elderly. Given the association between perceived neighbourhood disadvantage and residence in a rental flat block in our population, facilities for recreation in proximity to rental flat blocks can be enhanced in order to encourage exercise participation. There is less data, however, on the association between perceived neighbourhood quality and health screening participation, with a lack of previous studies investigating such an association in the published literature. A 2016 study demonstrated an association between perceptions of social and physical disorder, such as fear of crime and visible garbage, and cancer screening rates. Various reasons were postulated to support this association; perhaps the inherent disorder in lower-SES environments contributes to feelings of powerlessness and inaction regarding individuals’ own health, or the lack of social support and normative behaviors regarding screening in lower-SES environments contributes to low screening rates. In our study, we demonstrated an association between perceived poorer neighbourhood quality and lower odds of regular diabetes screening participation amongst non-diabetics. Given the known correlation between the built environment and various cardiovascular disease outcomes, poorer-quality neighbourhoods can be prioritized in cardiovascular screening interventions in order to facilitate early detection and mitigate the risk of cardiovascular disease.”
Reviewer 2 Report
One general remark, the title claims that the paper is focused on the analysis of the relation between the perception of the neighbourhood and health-related behaviours, but this aspect is very weakly included in the Methods, Results, and Discussion.
Methods and Results
The description of the methodology lacks details of how logistic regression models were developed:
- it is not clear what the threshold was used for dichotomization of the three constructs resulting from the NEWS-A scale
- how were regression models developed?
- what are general statistics for developed models?
- it is not obvious what independent variables were included in multivariate models for overall perceptions of neighbourhood environment and three components resulting from principal component analysis, did all variables the Tables 1 were included in the multivariate models for perception of neighbourhood?
- the same as above applies to the models developed for health-related behaviours as dependent variables
- it looks like only variables having a significant Impact on dependent variables are reported, why?
- furthermore, it is not clear if the multicollinearity was checked for all models
- it would be great to see what sociodemographic variables were included in regression models,
- was income also included in all models?
- in the tables providing the input from multivariate regression analysis all other variables apart from the perception of neighbourhood were classified as sociodemographic ones, which is hardly understandable for nor participation in community activities
- It looks that people below 75 years participated less in regular exercise; has respondents in this group were still professionally active?, if yes, what was their number.
What was the reliability assessment of the part of the NEWS-A scale used in the study? Is still valid if some part of the original instrument were used only?
The tables reporting the results of the statistical analysis still require clean-up, e.g. in the table 2, the ”married” option is repeated it occurs both under marital status and social isolation. The organisation of the independent variables in other tables is also peculiar – table 4: other sociodemographic variables including actually not such variables, instead dyslipidaemia, medical follow-up and self-assessment of health.
Discussion
The title of the paper is focused on meaning of the perception of the neighbourhood for undertaking health-related behaviour, but in the Discussion this topic is actually neglected and limited to one small paragraph. One could expect much more considerations about obtained results in this context.
The Authors tend to conclude the perception of the neighbourhood environment can be treated as an independent variable for health-related behaviours. But earlier, they demonstrate that many determinants of the such behaviours have also relations with the perception as assessed with the NEWS-A scale. More extensive discussion of possible factors not included in the their analysis and somehow reflected by the NEWS-A scale influencing health-related behaviours would be needed.
Author Response
Reviewer 2:
One general remark, the title claims that the paper is focused on the analysis of the relation between the perception of the neighbourhood and health-related behaviours, but this aspect is very weakly included in the Methods, Results, and Discussion.
We thank the reviewer for the feedback. We have sought to augment this in the Discussion, by including the following lines:
“Apart from physical characteristics of the neighbourhood environment that may promote walkability, better perceived neighborhood quality may also promote an improved sense of purpose and meaning in life, which may predispose individuals to take control of their own health, such as increasing exercise frequency and physical activity. Poorer perceived neighbourhood quality may also contribute to increased loneliness amongst community-dwelling elderly; which can reduce physical activity, given the importance of social presence in encouraging physical activity amongst the elderly. Given the association between perceived neighbourhood disadvantage and residence in a rental flat block in our population, facilities for recreation in proximity to rental flat blocks can be enhanced in order to encourage exercise participation. There is less data, however, on the association between perceived neighbourhood quality and health screening participation, with a lack of previous studies investigating such an association in the published literature. A 2016 study demonstrated an association between perceptions of social and physical disorder, such as fear of crime and visible garbage, and cancer screening rates. Various reasons were postulated to support this association; perhaps the inherent disorder in lower-SES environments contributes to feelings of powerlessness and inaction regarding individuals’ own health, or the lack of social support and normative behaviors regarding screening in lower-SES environments contributes to low screening rates. In our study, we demonstrated an association between perceived poorer neighbourhood quality and lower odds of regular diabetes screening participation amongst non-diabetics. Given the known correlation between the built environment and various cardiovascular disease outcomes, poorer-quality neighbourhoods can be prioritized in cardiovascular screening interventions in order to facilitate early detection and mitigate the risk of cardiovascular disease.”
Methods and Results
The description of the methodology lacks details of how logistic regression models were developed:
- it is not clear what the threshold was used for dichotomization of the three constructs resulting from the NEWS-A scale
We thank the reviewer for the feedback. We have provided information on the threshold used for dichotomization:
For Table 2, we include the following footnote:
“Factor analysis of the 17 NEWS items utilised in our study derived a total of 3 principal components, which were summarised as “perceived safety and convenience”, “perceived physical environment”, and “perceived proximity to recreational areas”. The component subscales for each principal component were summated and the results for each principal factor were dichotomised using the median result as the cut-off. The median score for “perceived safety and convenience” was 30 (min=12, max=45). The median score for “perceived physical environment” was 20 (min=4, max=16). The median score for “perceived proximity for recreational areas” was 3 (min=1, max=5).”
- how were regression models developed?
We thank the reviewer for the query. We have tried to increase the clarity by including the following lines in the Methods,
“For multivariate analysis, the most parsimonious logistic regression model was constructed by using a criterion of p-value<0.1 on univariate analysis as a cut-off for entry of factors into the final multivariate model; and removing non-significant variables in a stepwise fashion till the most parsimonious model was achieved.”
We have also included the following footnote in Table 3 (previous Table 2) and Table 5 (previous Table 4).
“The most parsimonious logistic regression model was constructed by using a criterion of p-value<0.1 on univariate analysis as a cut-off for entry of factors into the final multivariate model; and removing non-significant variables in a stepwise fashion till the most parsimonious model was achieved. All variables significant on multivariate analysis are enumerated.”
- what are general statistics for developed models?
We thank the reviewer for the query. We have tried to increase the clarity by including the calculated R2 of all developed models as a footnote in each relevant Table (Table 3, Table 5).
- it is not obvious what independent variables were included in multivariate models for overall perceptions of neighbourhood environment and three components resulting from principal component analysis, did all variables the Tables 1 were included in the multivariate models for perception of neighbourhood?
- the same as above applies to the models developed for health-related behaviours as dependent variables
We thank the reviewer for the query. We have tried to increase the clarity by including the following lines in the Methods,
“For multivariate analysis, the most parsimonious logistic regression model was constructed by using a criterion of p-value<0.1 on univariate analysis as a cut-off for entry of factors into the final multivariate model; and removing non-significant variables in a stepwise fashion till the most parsimonious model was achieved.”
We have also included the following footnote in Table 3 (previous Table 2) and Table 5 (previous Table 4).
“The most parsimonious logistic regression model was constructed by using a criterion of p-value<0.1 on univariate analysis as a cut-off for entry of factors into the final multivariate model; and removing non-significant variables in a stepwise fashion till the most parsimonious model was achieved. All variables significant on multivariate analysis are enumerated.”
- it looks like only variables having a significant Impact on dependent variables are reported, why?
We thank the reviewer for the opportunity to clarify. As we presented the most parsimonious model, only variables having a significant impact on dependent variables were included in the final multivariate model. We have attempted to clarify this by making the following changes as outlined above.
- furthermore, it is not clear if the multicollinearity was checked for all models
We thank the reviewer for the opportunity to clarify. We did check multicollinearity and no multicollinearity was detected. We have clarified this in the Methods: “Variables included in final multivariate logistic regression models were also evaluated for multicollinearity, with a view to removing redundant variables; however, no multicollinearity was detected.”
- it would be great to see what sociodemographic variables were included in regression models,
We thank the reviewer for the opportunity to clarify. As we presented the most parsimonious model, only sociodemographic variables having a significant impact on dependent variables were included in the final multivariate model.
- was income also included in all models?
We thank the reviewer for the opportunity to clarify. As we presented the most parsimonious model, only sociodemographic variables having a significant impact on dependent variables were included in the final multivariate model. Hence, income was not included in all models.
- in the tables providing the input from multivariate regression analysis all other variables apart from the perception of neighbourhood were classified as sociodemographic ones, which is hardly understandable for nor participation in community activities
We thank the reviewer for the feedback. We have reorganized the format of Table 5 to divide into demographic, medical and social factors, where relevant, similar to the presentation in Table 1 and Table 2. We hope this makes it clearer.
- It looks that people below 75 years participated less in regular exercise; has respondents in this group were still professionally active?, if yes, what was their number.
Yes, some of the respondents in this group would still be professionally active. The overall employment rate was 45% (238/528).
What was the reliability assessment of the part of the NEWS-A scale used in the study?
We thank the reviewer for the feedback. We have included the following line in the Methods:
“The Cronbach-alpha coefficient for the subscale items of the modified NEWS-A score was 0.71, suggesting reasonable internal consistency.”
Is still valid if some part of the original instrument were used only?
While there was reasonable internal consistency with the modified NEWS-A score, we agree with the reviewer that partial usage of the NEWS-A score may affect validity. We have included the following statement in the Limitations: “Finally, while elements of the NEWS-A score have previously been used to evaluate perceptions of the built environment in our local population, partial usage of the NEWS-A score may potentially affect the validity of our measurements.”
The tables reporting the results of the statistical analysis still require clean-up, e.g. in the table 2, the ”married” option is repeated it occurs both under marital status and social isolation.
We have examined Table 3 (ex-Table 2) and marital status is a significant factor independently associated with overall perception of the neighborhood as more disadvantaged as well as less safe/convenient, hence it is repeated twice as it appears in two separate multivariate logistic regression models. To make this clearer, we have standardized the order of presentation so that “marital status” is presented before “social isolation” in both multivariate regression models.
The organisation of the independent variables in other tables is also peculiar – table 4: other sociodemographic variables including actually not such variables, instead dyslipidaemia, medical follow-up and self-assessment of health.
We thank the reviewer for the feedback. We have reorganized the format of Table 5 to divide into demographic, medical and social factors,, instead of presenting it as just “other sociodemographic variables”where relevant, similar to the presentation in Table 1 and Table 2. We hope this makes it clearer.
Discussion
The title of the paper is focused on meaning of the perception of the neighbourhood for undertaking health-related behaviour, but in the Discussion this topic is actually neglected and limited to one small paragraph. One could expect much more considerations about obtained results in this context. The Authors tend to conclude the perception of the neighbourhood environment can be treated as an independent variable for health-related behaviours. But earlier, they demonstrate that many determinants of the such behaviours have also relations with the perception as assessed with the NEWS-A scale. More extensive discussion of possible factors not included in the their analysis and somehow reflected by the NEWS-A scale influencing health-related behaviours would be needed.
We note the Reviewer’s comment and thank the Reviewer for the feedback. We have augmented the Discussion as follows:
“Apart from physical characteristics of the neighbourhood environment that may promote walkability, better perceived neighborhood quality may also promote an improved sense of purpose and meaning in life, which may predispose individuals to take control of their own health, such as increasing exercise frequency and physical activity. Poorer perceived neighbourhood quality may also contribute to increased loneliness amongst community-dwelling elderly; which can reduce physical activity, given the importance of social presence in encouraging physical activity amongst the elderly. Given the association between perceived neighbourhood disadvantage and residence in a rental flat block in our population, facilities for recreation in proximity to rental flat blocks can be enhanced in order to encourage exercise participation. There is less data, however, on the association between perceived neighbourhood quality and health screening participation, with a lack of previous studies investigating such an association in the published literature. A 2016 study demonstrated an association between perceptions of social and physical disorder, such as fear of crime and visible garbage, and cancer screening rates. Various reasons were postulated to support this association; perhaps the inherent disorder in lower-SES environments contributes to feelings of powerlessness and inaction regarding individuals’ own health, or the lack of social support and normative behaviors regarding screening in lower-SES environments contributes to low screening rates. In our study, we demonstrated an association between perceived poorer neighbourhood quality and lower odds of regular diabetes screening participation amongst non-diabetics. Given the known correlation between the built environment and various cardiovascular disease outcomes, poorer-quality neighbourhoods can be prioritized in cardiovascular screening interventions in order to facilitate early detection and mitigate the risk of cardiovascular disease.”
Reviewer 3 Report
In the manuscript "Perceived neighbourhood environment and its association with health behaviours amongst low-income public rental flat residents in Singapore" the Authors evaluated factors associated with perceptions of the neighbourhood environment and its association with health-seeking behaviors.
In my opinion the study is very interesting and well written. Only some minor revision should be made.
At line 112 you refer to standardized questionnaires: please insert the citation related to the standardization or, alternatively, indicate how the standardization was carried out.
Reference no. 16 should be re-write as follows: Liang En, W., Sin, D., Wen Qi, C., Zong Chen, L., Shibli, S., Choon-Huat Koh, G. Chronic pain in a low socioeconomic status population in Singapore: a cross-sectional study. Pain Medicine, 2016,17(5), 864-876.
Please correct the wrapped text at lines 394-395.
Overall, in the text, reference numbers should be placed in square brackets [ ], and placed before the punctuation; for example [1], [1–3] or [1,3].
All references should be edited as reported in the Microsoft Word template. You can find the template in the instruction for authors.
Tables should be placed in the main text near to the first time they are cited and edited as reported in the Microsoft Word template.
Author Response
Reviewer 3:
In the manuscript "Perceived neighbourhood environment and its association with health behaviours amongst low-income public rental flat residents in Singapore" the Authors evaluated factors associated with perceptions of the neighbourhood environment and its association with health-seeking behaviors.
In my opinion the study is very interesting and well written. Only some minor revision should be made.
We thank the reviewer for the feedback.
At line 112 you refer to standardized questionnaires: please insert the citation related to the standardization or, alternatively, indicate how the standardization was carried out.
We have amended the relevant lines to read,
“At baseline, information on residents’ sociodemographic characteristics, medical, functional and social status was collected via interviewer-administered standardized questionnaires in English, Chinese and Malay. Standardised training for questionnaire administration was provided to interviewers by the study coauthors.”
Reference no. 16 should be re-write as follows: Liang En, W., Sin, D., Wen Qi, C., Zong Chen, L., Shibli, S., Choon-Huat Koh, G. Chronic pain in a low socioeconomic status population in Singapore: a cross-sectional study. Pain Medicine, 2016,17(5), 864-876.
We thank the reviewer for the feedback. We have done so.
Please correct the wrapped text at lines 394-395.
We thank the reviewer for the feedback. We have done so.
Overall, in the text, reference numbers should be placed in square brackets [ ], and placed before the punctuation; for example [1], [1–3] or [1,3].
We thank the reviewer for the feedback. We have done so.
All references should be edited as reported in the Microsoft Word template. You can find the template in the instruction for authors.
We thank the reviewer for the feedback. We have done so.
Tables should be placed in the main text near to the first time they are cited and edited as reported in the Microsoft Word template.
We thank the reviewer for the feedback. We have done so.
Round 2
Reviewer 2 Report
I must confirm once again, although the title of the paper promises the input on the association of neighbourhood quality with health behaviours, this aspect is marginal in the paper. Even in the Abstract in the Results the Authors mention only about reduced exercise and screening participation, but they do not provide relevant statistics. If health behaviours are a priority for the paper, why they are not accompanied by OR and 95%CI? But actually, among these preventative services the impact of the neighbourhood was significant only in case of regular diabetes screening and none of 4 other. So, are we really able to say that neighbourhood perception is related to screening practices. All these issues should be clearly stated in the Abstract.
Overall, it looks like the Authors extracted three subscales from the NEWS scale and then combined relevant items to make a new scale which was examined for main components once again. In the result, they obtained different three components than those three included initially in the analysis. I’m not sure if, after such manipulations, a new scale can be called NEWS at all. And a new scale would need full psychometric evaluation to confirm its validity and reliability.
One can understand that multivariate „the most parsimonious” regression models are specified in Table 3. But still, it is not clear what the Authors mean in their explanations about the double appearance of the variable „married” in table 3 in the relations to the model for „Overall neighbourhood perception”. Even for people performing similar types of statistical analysis, it is a source of confusion, not mentioning less prepared readers. If the authors somehow developed two models for „overall …”, they should be specified separately.
Conclusions, as stated in the Abstract and relevant part of the manuscript, do not reflect the results in the part on the preventative services. A more nuanced statement is required.
On the side of the appearance of tables, now they are formatted in a very peculiar way, in the result, it is rather difficult to understand which independent belong to which regression model (I mean the width of horizontal lines in the tables)
Author Response
Reviewer 2:
I must confirm once again, although the title of the paper promises the input on the association of neighbourhood quality with health behaviours, this aspect is marginal in the paper.
We thank the reviewer for the feedback and suggestions on how to improve our paper.
We agree with the reviewer that the main thrust of the paper was regarding the factors associated with perceived neighborhood quality and its association with health screening and exercise participation, specifically. There is a lack of studies on perceived neighborhood quality in Asian societies, and its link to health, as we alluded to in the Introduction; in our local setting, few studies have investigated perceived neighborhood quality and its association with health behaviours. However, we do agree with the reviewer that we only present information on health screening and exercise participation, but the range of other health behaviors is much wider and the paper title as it currently stands can imply association beyond the health behaviors that we actually studied and discussed.
As such, we amend our paper title to “Perceived neighbourhood environment and its association with health screening and exercise participation amongst low-income public rental flat residents in Singapore.”
We hope that this improves the clarity for the reader, and specifically highlights that we only investigated the relationship between neighbourhood quality and health screening/exercise participation.
Even in the Abstract in the Results the Authors mention only about reduced exercise and screening participation, but they do not provide relevant statistics. If health behaviours are a priority for the paper, why they are not accompanied by OR and 95%CI?
We thank the reviewer for the feedback.
We have amended the abstract and replaced the generic “health behaviours” with “participation in regular exercise and health screening” throughout the Abstract, so as to increase the clarity of the specific health behaviours that we refer to.
As per the reviewer’s suggestion, we have also included the specific OR and 95%CI for the associations between health screening and exercise and perceived neighborhood quality. These were previously omitted because of word count- however we have reduced words in other areas of the Abstract to include this.
But actually, among these preventative services the impact of the neighbourhood was significant only in case of regular diabetes screening and none of 4 other. So, are we really able to say that neighbourhood perception is related to screening practices. All these issues should be clearly stated in the Abstract.
We thank the reviewer for the feedback.
To be specific and clear, in our Results section of the Abstract, we have amended the wording to be clear that we are referring to diabetes screening alone:
“Perceptions of neighbourhood disadvantage were independently associated with reduced exercise participation (aOR=0.67, 95%CI=0.45-0.98) and reduced screening participation in diabetes screening (aOR=0.63, 95%CI=0.41-0.95).”
In the Conclusions section of the abstract, we have also amended the wording to be clear that we are referring to diabetes screening alone:
“Perceived neighbourhood disadvantage associates with lower participation in regular exercise and diabetes screening.”
Overall, it looks like the Authors extracted three subscales from the NEWS scale and then combined relevant items to make a new scale which was examined for main components once again. In the result, they obtained different three components than those three included initially in the analysis. I’m not sure if, after such manipulations, a new scale can be called NEWS at all.
We thank the reviewer for the feedback.
We do agree that only 3 subscales of the NEWS-A were used in this analysis. Previous usage of the NEWS score in our local setting also used selected subscales rather than the entire 64-item NEWS-A; taking into account the subscales that were considered to be relevant for their purposes.
We allude to this in the Methods section, “In its original form, the NEWS-A comprised a total of 64 items spread across 12 subscales. However, not all subscales were relevant to the local context. The NEWS has been previously modified for use in the Singaporean setting [32], preserving 8 subscales of residential density, land use mix (diversity), land use mix (access), street connectivity, infrastructure (places for walking and cycling), aesthetics (neighbourhood surroundings), traffic safety and safety from crime. It has previously been utilised in the Singaporean context to study the impact of the built environment on physical activity [32]; however, in a separate study, specific subscales of the NEWS were utilised in analysing the impact of the built environment on cognitive impairment in a population of Singaporean community-dwelling elderly [33].”
We do agree with the reviewer that the scale has been modified. As such, we have revised our terminology throughout the paper, to highlight that this study specifically utilised the NEWS-A subscales of crime safety, land-use mix (access) and land-use mix (diversity) but omitted the other subscales.
This is to improve clarity and highlight that we only used a subset of the NEWS and avoid giving the inadvertent assumption that we used the full NEWS when this was not the case.
Examples of these changes include:
In Statistical analysis:
“Factor analysis of the 17 NEWS items (from the NEWS-A subscales of crime safety, land use access and land use diversity) utilised in our study was conducted”
In Results:
“On factor analysis, the 17 items composing the NEWS score (initially obtained from the NEWS-A subscales of crime safety, land use access and land use diversity) could be reduced to 3 principal components”
In Table 1, footnote 1:
“The median total score derived from the NEWS-A subscales of crime safety, land use access and land use diversity (with a maximum score of 76 and a minimum score of 17) was used as a cutoff to dichotomise into “overall neighbourhood perception: less disadvantaged” and “overall neighbourhood perception: more disadvantaged”. The median score was 40 (interquartile ratio=28-60).”
In Table 2, footnote 1:
“Factor analysis of the 17 NEWS items from the subscales of crime safety, land use access and land use diversity utilised in our study derived a total of 3 principal components, which were summarised as “perceived safety and convenience”, “perceived physical environment”, and “perceived proximity to recreational areas”. The component subscales for each principal component were summated and the results for each principal factor were dichotomised using the median result as the cut-off. The median score for “perceived safety and convenience” was 30 (min=12, max=45). The median score for “perceived physical environment” was 20 (min=4, max=16). The median score for “perceived proximity for recreational areas” was 3 (min=1, max=5).”
We also specify in the Limitations that we only used a specific subset of subscales from the NEWS-A and that this affects the validity of our conclusions as well as generalizability. We have further elaborated on it, by modifying the following sentences:
“Finally, we elected to use subscales of the NEWS-A in our assessment of perceived neighborhood quality that were relevant to our study context, similar to previous studies in our local population that also utilized selected elements of the NEWS-A [32-33]; the rationale was that previous local studies could serve as a potential basis of comparison, given the extremely limited number of local studies that have evaluated neighbourhood perceptions. However, we acknowledge that partial usage of the NEWS-A questionnaire, which was specifically designed to evaluate perceptions of neighbourhood walkability, may potentially affect the validity of our measurements
And a new scale would need full psychometric evaluation to confirm its validity and reliability.
We thank the reviewer for the feedback and appreciate it.
Previously, we utilized limited subscales of the NEWS-A (instead of the full 64 item questionnaire) as mentioned in our Methodology. The NEWS-A was chosen as it is one of the only questionnaires that has been utilized to study neighbourhood perceptions in our local context, and we were keen to use it so as to provide a basis of comparison to other local studies that considered neighbourhood perceptions. However, other studies also utilized selected subscales, not the full NEWS-A score, given that the NEWS-A score was also designed specifically to assess neighbourhood perceptions with regards to walkability. We recognize, though, the concern that using an abbreviated scale may affect the reliability of our measurements. Given concerns regarding usage of an abbreviated scale, in addition to using the total summated score of the NEWS-A subscales that we utilised, we also conducted factor analysis to identify the principal factors and assessed internal consistency by evaluating the Cronbach-alpha statistic. We do acknowledge the limitations of this approach, and recognize that future research directions would be served by formulating a instrument specifically designed to assess neighbourhood perceptions in our local context, perhaps by utilizing some of the key dimensions of neighbourhood quality (eg. perceived safety/convenience and perceived physical environment) that were identified in this study. We have emphasized this in the Limitations section:
“Finally, we elected to use subscales of the NEWS-A in our assessment of perceived neighborhood quality that were relevant to our study context, similar to previous studies in our local population that also utilized selected elements of the NEWS-A [32-33]; the rationale was that previous local studies could serve as a potential basis of comparison, given the extremely limited number of local studies that have evaluated neighbourhood perceptions. However, we acknowledge that partial usage of the NEWS-A questionnaire, which was specifically designed to evaluate perceptions of neighbourhood walkability, may potentially affect the validity of our measurements; elements of perceived neighbourhood quality identified in this study (safety and convenience; physical environment) may form the basis for future constructs designed to specifically assess neighborhood perceptions in our local population.”
One can understand that multivariate „the most parsimonious” regression models are specified in Table 3. But still, it is not clear what the Authors mean in their explanations about the doubleappearance of the variable „married” in table 3 in the relations to the model for „Overall neighbourhood perception”. Even for people performing similar types of statistical analysis, it is a source of confusion, not mentioning less prepared readers. If the authors somehow developed two models for „overall …”, they should be specified separately.
We thank the reviewer for the feedback.
We apologise- there was a typo in the table and hence the presence of an additional row in Table 3, (“Married: 0.58 (0.39-0.88)”,which is erroneous and should be removed. The correct value is the first value for marital status (“Married: 0.49 (0.33-0.73)”) which occurred earlier in the table. We apologise for not realizing this double appearance earlier and thank the reviewer for pickling it up. We have removed the erroneous row in question.
Conclusions, as stated in the Abstract and relevant part of the manuscript, do not reflect the results in the part on the preventative services. A more nuanced statement is required.
We thank the reviewer for the feedback.
We agree with the reviewer that although we looked at cancer and cardiovascular screening participation, we only identified a significant association between perceived neighbourhood quality and diabetes screening, with no significant association identified for gynecological cancer screening. We had only discussed diabetes screening as that was the only significant association; however, we agree with the reviewer that our findings for cancer screening and hypertension/dyslipidemia screening should also be addressed in the Discussion for completeness. We hypothesise that the differences for cancer screening/hypertension/dyslipidemia may be due to differences in the way in which screening is carried out, with only diabetes/dyslipidemia screening being carried out at primary care clinics within the immediate neighbourhood (and hence possibly resulting in an association between perceived neighbourhood quality and diabetes screening participation, but not for other modalities).
As such, we have incorporated the following paragraph in the Discussion:
“In our study, although there was no significant association between perceived neighborhood quality and gynecological cancer screening, we demonstrated an association between perceived poorer neighbourhood quality and lower odds of regular diabetes screening participation amongst non-diabetics. There were lower proportions of residents participating in regular dyslipidemia screening amongst those staying in areas with poorer perceived neighbourhood quality; conversely, higher proportions of residents staying in areas with poorer perceived neighbourhood quality participated in regular hypertension screening, although in both cases the association was not statistically significant. In Singapore, while residents in rental flat blocks have lower rates of participation in regular cancer and cardiovascular screening [21-22], there are significant differences in accessibility between screening modalities. In general, cardiovascular screening takes place at either private general practitioners (GPs) or public primary care clinics (“polyclinics”) in the immediate neighbourhood; [21] these clinics are distributed at relatively high densities, with at least one clinic within a 0.5 kilometre radius of the residential block. [23] However, in the case of cancer screening, subsidized screening is generally available only at designated polyclinics with the necessary facilities (eg. imaging); [22] hence the facilities providing subsidized cancer screening are generally located outside of the immediate neighbourhood. This may account for why we detected a significant association for diabetes screening, but not cancer screening; as diabetes screening occurs at centres within the immediate neighborhood, perceived neighbourhood quality does impact on perceived accessibility and hence diabetes screening participation, whereas cancer screening, occurring outside the immediate neighbourhood, is less impacted by the perceived quality of the immediate neighbourhood. Hypertension screening, unlike diabetes screening, was higher in neighbourhoods with lower perceived neighbourhood quality, though the association was not statistically significant. Though hypertension management tends to be poorer in rental flat blocks, with prioritization of patients staying in rental housing for various interventions to improve chronic disease management; [15] with wider availability of home-based blood pressure monitoring and more widespread provision of blood pressure screening in door-to-door screening, [15] residents may no longer be so dependent on primary care clinics in the neighborhood for hypertension screening, compared with diabetes screening which still requires phlebotomy equipment for blood draws and a follow-up system to recall the patient to review results. Given the known correlation between the built environment and various cardiovascular disease outcomes [48-49], poorer-quality neighbourhoods can be prioritized in cardiovascular screening interventions in order to facilitate early detection and mitigate the risk of cardiovascular disease [21].”
In the conclusion, we have also amended it to be more specific:
“Having a poorer perception of the neighbourhood environment is associated with reduced participation in regular exercise and diabetes screening”
On the side of the appearance of tables, now they are formatted in a very peculiar way, in the result, it is rather difficult to understand which independent belong to which regression model (I mean the width of horizontal lines in the tables)
We thank the reviewer for the feedback.
For the tables that include regression models (specifically Table 3 and 5), we have attempted to format the tables to make this clearer, specifically by including a change in the greyscale contrast between the different regression models